# P2X1 and P2X7 Receptor Overexpression Is a Negative Predictor of Survival in Muscle-Invasive Bladder Cancer

**DOI:** 10.3390/cancers15082321

**Published:** 2023-04-16

**Authors:** Stephan Ledderose, Severin Rodler, Lennert Eismann, Georg Ledderose, Martina Rudelius, Wolfgang G. Junger, Carola Ledderose

**Affiliations:** 1Institute of Pathology, Ludwig Maximilian University, 80337 Munich, Germany; 2Department of Urology, Ludwig Maximilian University, 81377 Munich, Germany; 3Department of Oto-Rhino-Laryngology, Ludwig Maximilian University, 81377 Munich, Germany; 4Department of Surgery, Beth Israel Deaconess Medical Center, Harvard Medical School, Boston, MA 02215, USA; 5Department of Surgery, University of California San Diego Health, La Jolla, CA 92037, USA

**Keywords:** bladder cancer, muscle-invasive bladder cancer, ATP, purinergic signaling, P2X receptors, P2X1 receptor, P2X7 receptor, biomarker, prognosis, T24 cells

## Abstract

**Simple Summary:**

Bladder cancer is one of the most common malignancies. The prognosis is particularly poor for advanced cancer. There is a need for new prognostic markers to guide treatment decisions and promote the development of novel therapeutic strategies to increase the success rate of current treatment regimens. The tumor microenvironment is characterized by an increase in extracellular ATP levels. ATP-recognizing P2X receptors have been implicated in the growth and metastasis of various malignant cancers. Here, we analyzed the potential of different P2X receptor subtypes as prognostic markers in muscle-invasive bladder cancer. In vitro experiments confirmed a growth-promoting effect of extracellular ATP and P2X receptors on bladder cancer cells. In agreement, our analyses of cancer tissue samples from 173 patients showed that expression of P2X1 and P2X7 receptors is an independent negative predictor of survival and a potential therapeutic target in muscle-invasive bladder cancer.

**Abstract:**

Bladder cancer is amongst the most common causes of cancer death worldwide. Muscle-invasive bladder cancer (MIBC) bears a particularly poor prognosis. Overexpression of purinergic P2X receptors (P2XRs) has been associated with worse outcome in several malignant tumors. Here, we investigated the role of P2XRs in bladder cancer cell proliferation in vitro and the prognostic value of P2XR expression in MIBC patients. Cell culture experiments with T24, RT4, and non-transformed TRT-HU-1 cells revealed a link between high ATP concentrations in the cell culture supernatants of bladder cell lines and a higher grade of malignancy. Furthermore, proliferation of highly malignant T24 bladder cancer cells depended on autocrine signaling through P2X receptors. P2X1R, P2X4R, and P2X7R expression was immunohistochemically analyzed in tumor specimens from 173 patients with MIBC. High P2X1R expression was associated with pathological parameters of disease progression and reduced survival time. High combined expression of P2X1R and P2X7R increased the risk of distant metastasis and was an independent negative predictor of overall and tumor-specific survival in multivariate analyses. Our results suggest that P2X1R/P2X7R expression scores are powerful negative prognostic markers in MIBC patients and that P2XR-mediated pathways are potential targets for novel therapeutic strategies in bladder cancer.

## 1. Introduction

Urinary bladder cancer accounts for about 573,000 newly diagnosed cancer cases and 212,000 deaths worldwide each year [1]. Approximately 25% of new bladder cancer cases are detrusor muscle-invasive bladder cancers (MIBC; ≥pT2), and about 10% are expanding beyond the bladder muscle wall into perivesical tissue or adjacent organs or have already metastasized [2,3]. The recommended first-line treatment of MIBC is radical cystectomy with pelvic lymph node dissection. However, about 50% of patients experience recurrence and develop metastatic disease [2,4].

While cisplatin-based chemotherapy remains the standard therapy perioperatively and in patients with metastatic disease, several new agents have been introduced for the treatment of bladder cancer over the last decade, including immune checkpoint inhibitors and targeted therapies with antibody–drug conjugates [2,5]. Nevertheless, the prognosis of advanced bladder cancer remains poor with a 5-year overall survival rate of 5–30% [2], and existing treatment options are associated with severe side effects and a significant reduction in the quality of life [3,4,5]. Thus, there is a need for new reliable prognostic markers to facilitate individualized treatment decisions and inspire new therapeutic approaches.

The tumor microenvironment (TME) plays a critical role in tumor growth and progression [6]. The purine nucleotide adenosine 5’-triphosphate (ATP) is a major component of the TME [7,8]. ATP can be actively secreted by tumor cells or host cells or passively released after cell damage or stress [8]. Besides its role as the universal energy carrier for all cellular reactions, ATP functions as an extracellular messenger molecule through the activation of nucleotide receptors of the P2X and P2Y subfamilies [9]. ATP is the natural ligand of the seven P2X receptors (P2XR1-7), which are cation channels, while the eight known P2Y receptors (P2YR1,2,4,6,11–14) couple to G-proteins and recognize ATP and other nucleotides like ADP or UTP [10]. Ectonucleotidases stepwise convert extracellular ATP to adenosine, the ligand of the four P1 or adenosine receptors [11]. Virtually all cell types and tissues express different sets of purinergic receptors. Accordingly, purinergic signaling is involved in the regulation of numerous cell functions such as apoptosis, proliferation, migration, and cell differentiation [10,12,13]. In the lower urinary tract, stretch-induced ATP release from urothelial cells and stimulation of purinergic receptors is involved in bladder contraction and relaxation [14]. Alterations in purinergic signaling and purinergic receptor expression have been implicated in different pathological conditions of the bladder, including interstitial cystitis, overactive bladder syndrome, and idiopathic detrusor instability [14,15,16].

The accumulation of ATP in the TME is often paralleled by altered P2 receptor expression patterns in tumor tissues [17]. Changes in the purinergic tumor environment can promote or inhibit tumor growth, depending on tumor type, extracellular nucleotide and nucleoside concentrations, and the expression levels and distribution patterns of purinergic receptor subtypes in tumor and immune cells [8,17]. P2XR overexpression has been associated with tumor growth and progression in different cancer types [8]. For instance, P2X7R and P2X4R activation enhances the invasiveness of prostate cancer [18], and P2X1R and P2X7R signaling promotes the proliferation of leukemia cells [19]. Expression profiles of different P2 receptors were described as independent prognostic factors in a variety of malignant tumor entities, including cervical, lung, liver, gastric and renal cell carcinoma [20,21,22,23,24,25].

The purpose of this study was to investigate the role of P2XRs in bladder cancer cell proliferation and the prognostic value of P2XR expression in MIBC. We focused on P2X1R, P2X4R, and P2X7R as these receptors have been repeatedly associated with tumor growth in different cancers [18,19,26] and are also the predominant P2XRs expressed by immune cells [27]. We found that extracellular ATP and P2XRs promote the growth of T24 bladder cancer cells in vitro and that high P2X1R expression alone or in combination with high P2X7R expression is associated with reduced disease-free survival (DFS), tumor-specific survival (TSS), and overall survival (OS) in patients with MIBC.

## 2. Materials and Methods

### 2.1. Patients

This retrospective study was approved by the Medical Ethics Committee of Ludwig Maximilian University (LMU) Munich (reference number 20–179). The study group consisted of 173 patients who were diagnosed with MIBC and underwent radical cystectomy at the Department of Urology (LMU Munich) between 2004 and 2014. Surgery was performed with curative or palliative intent using standardized procedures for urinary diversion by ileal conduit or ileal neobladder formation. All histological specimens were systematically reviewed to confirm tumor type, grade, and stage by experienced pathologists. Staging was performed according to the AJCC/UICC TNM staging guidelines (8th edition) and the latest WHO classification of genitourinary tumors (5th edition) [28,29]. Patients who had received neoadjuvant therapy or intravesical therapy with Bacillus Calmette-Guérin or mitomycin prior to surgery were excluded. Other exclusion criteria were incompletely archived material or incomplete medical history, low-grade tumors, and cancer types that did not fulfill the histopathological criteria for urothelial carcinoma and its histological subtypes as defined by the WHO 2022 classification [29]. Follow-up was performed according to the European Association of Urology guidelines [5]. Patient characteristics, clinicopathological parameters, DFS, TSS, and OS were documented and analyzed for associations with P2X1R, P2X4R, and P2X7R expression.

### 2.2. Tissue Microarray Construction

Expression of P2X1R, P2X4R, and P2X7R was examined in tissue microarrays (TMAs) using formalin-fixed, paraffin-embedded (FFPE) tissue blocks from 173 MIBCs. Punched-out 1 mm tumor cores from three different tumor areas per patient were arrayed into new TMA blocks. Triplicates of each sample were used to minimize tissue loss and to overcome tumor heterogeneity. The embedded tissue was cut into 4 µm thick slices and used for immunohistochemical analyses.

### 2.3. Immunohistochemistry

Antibody staining was established with appropriate isotype and system controls. Tonsil tissue was used as a positive control and was included in each staining run. All antibodies had been validated by the manufacturers against Western blot and RNA-seq data on a wide range of normal and cancer tissue.

For P2X1R staining, antigen retrieval was carried out by heat treatment using the antigen retrieval AR-10 kit (DCS, Hamburg, Germany; HK057-5KE). Slides were incubated with the polyclonal rabbit anti-human P2X1R primary antibody (1:300; Abcam, Berlin, Germany; ab74058) for 60 min at room temperature. Bound antibodies were detected by the use of the ImmPRESS Anti-Rabbit IgG Polymer Kit (Vector Laboratories, Newark, CA, USA; MP-7401).

For P2X4R staining, slides were subjected to heat pre-treatment with ProTaqs IV Antigen Enhancer (Quartett, Berlin, Germany; 401602392). After incubation with the rabbit polyclonal anti-human P2X4R primary antibody (1:80; Atlas Antibodies, Lund, Sweden; HPA039494) for 60 min at room temperature, the MACH 3 Rabbit HRP Polymer Detection kit (Biocare, Pacheco, CA, USA; M3R531) was used as a detection system.

For P2X7R staining, slides were subjected to heat retrieval with Novocastra Epitope Retrieval Solution pH 8.0 (Leica Biosystems, Wetzlar, Germany; RE7116). After incubation with the monoclonal mouse anti-human P2X7R antibody (1:250; Atlas Antibodies; AMAb91714) for 60 min at room temperature, bound antibodies were detected with the MACH 3 Mouse HRP Polymer Detection system (Biocare; M3M530).

All reactions were visualized using DAB+ (Agilent Technologies, Santa Clara, CA, USA; K3468) as a chromogen. Sections were then counterstained with hematoxylin (Vector Laboratories; H-3401), dehydrated, and mounted.

### 2.4. Semiquantitative Analysis of P2X Receptor Expression

Immunoreactivity of P2XRs was scored using the histochemical scoring system (H-score) as previously described [30]. The H-score incorporates both the staining intensity and the percentage of stained cells at each intensity level. Intensity was scored as 0 (no evidence of staining), 1 (weak staining), 2 (moderate staining), and 3 (strong staining). The final H-score is the sum of the intensity values multiplied by the percentage of stained cells. Three tumor cores per patient were scored separately and averaged to obtain the final H-score of each tumor sample. If a TMA core did not contain any tumor tissue, it was excluded from the calculations of the final score. In total, stained tumor core sections from 171 patients were available for evaluation of P2X1R and P2X4R expression, and tumor cores from 172 patients were stained and evaluated for P2X7R expression. The median H-score was used as a cutoff value to define groups with low and high P2XR expression (P2X1R: H-score ≥ 25; P2X4R: H-score ≥ 10; P2X7R: H-score ≥ 60). The H-score for the combined expression of P2X1R and P2X7R was calculated as the sum of the single scores of both receptors. For this analysis, tumor cores from 170 patients were available. Again, tumors with low or high combined expression of P2X1R and P2X7R were distinguished from each other based on the median H-score (H-score ≥ 85).

### 2.5. Cell Culture

T24, RT4, and TRT-HU-1 cell lines were kindly provided by Dr. Rosalyn M. Adams (Boston Children’s Hospital, Boston, MA, USA). T24 and RT4 cells were cultured in McCoy’s 5A medium (Invitrogen, Thermo Fisher Scientific, Waltham, MA, USA) supplemented with 10% heat-inactivated fetal bovine serum (FBS; Gibco, Thermo Fisher Scientific), 100 U/mL penicillin, and 100 µg/mL streptomycin (Gibco). The hTERT-immortalized human urothelial cell line TRT-HU-1 was maintained in Dulbecco’s modified Eagle medium (DMEM; Invitrogen, Thermo Fisher Scientific) supplemented with 15% non-heat-inactivated FBS (Invitrogen), MEM non-essential amino acids (Gibco), and 0.01% 1-thioglycerol (Sigma-Aldrich, St. Louis, MO, USA). Cells were cultured at 37 °C in a humidified atmosphere containing 5% CO_2_ and 95% air.

### 2.6. Carboxyfluorescein Succinimidyl Ester (CFSE)-Based Proliferation Assay

Proliferation of T24, RT4, and TRT-HU-1 cells was assessed using the CellTrace™ CFSE Cell Proliferation Kit (Invitrogen) as recommended by the manufacturer with a few modifications. Briefly, cells were detached by trypsin, washed with phosphate-buffered saline (PBS), and resuspended at 5 × 10^6^ cells/mL in CFSE staining solution (5 µM in PBS). After 15 min incubation at 37 °C, ten volumes of cell culture medium were added, and the cell mixture was incubated for another 5 min. Then, cells were pelleted by centrifugation, washed once with culture medium, and seeded at 15,000 cells per well in a 24-well cell culture plate. If indicated, antagonists of P2X1R (10 µM NF023; Tocris, Bristol, UK), P2X4R (10 µM 5-BDBD, Tocris), or P2X7R (10 µM A438079, Tocris), the P2 receptor antagonist suramin (100 µM; Sigma-Aldrich), the ATP-degrading enzyme apyrase (1 U/mL; Sigma-Aldrich), the ATP release blocker carbenoxolone (CBX; 20 µM; Sigma-Aldrich), or the natural P2XR ligand ATP (10 µM; Sigma-Aldrich) were added, and the cells were maintained in a final volume of 500 µL in 5% CO_2_/95% air at 37 °C. Final drug concentrations were chosen based on previous studies by us and others [18,19]. At the indicated times, cells were detached by trypsin and analyzed with a NovoCyte 3000 flow cytometer (Agilent). Single cells were identified by plotting FSC-A vs. FSC-H, and the CFSE median fluorescence intensity (MFI) of at least 8000 single cells was determined. Exponential growth curves were generated by plotting 1/MFI as a function of time [31]. The function N(t) = N(0) × e*^k^*^t^ was fitted to the data by non-linear regression (SigmaPlot 12.5, Systat Software Inc., San Jose, CA, USA) to determine the growth rate *k* and the doubling time, DT (DT = ln2/*k*).

### 2.7. ATP Concentrations in Cell Culture Supernatants

T24, RT4, and TRT-HU-1 cells were seeded in 24-well plates at a density of 3.75 × 10^5^ cells per well in a final volume of 500 µL and allowed to attach for 3 h in an incubator adjusted to 5% CO_2_/95% air at 37 °C. Cell culture supernatants were collected, cooled in an ice bath for 10 min, and centrifuged twice to ensure cell-free samples. Perchloric acid (400 mM) was added to the supernatants to stop any enzymatic activity. Then, samples were processed, and ATP concentrations were determined by high-performance liquid chromatography (HPLC) as previously described [32].

### 2.8. Statistics

Statistical analyses were performed with SigmaPlot 12.5 software. OS was defined as the time between primary surgery and death from any cause. For TSS, death caused by bladder cancer was defined as the clinical endpoint. Patients who were alive at the end of follow-up were censored. DFS refers to the time between primary surgery and relapse. Patients were censored for DFS if recurrent cancer and metastasis were absent at the end of follow-up or at the time of death. Survival curves were calculated using the Kaplan–Meier method and compared by the log-rank test. Associations between P2XR expression, demographic characteristics (age, sex), and clinicopathological parameters (pT staging category, pN stage, distant metastasis (M), lymphovascular invasion (L), blood vessel invasion (V), perineural invasion (Pn), resection margin (R), UICC stage, receipt of adjuvant therapy) were examined using the Chi-square test or unpaired t-test for categorical and numerical variables, respectively. To assess the prognostic value of low or high P2XR expression, multivariable hazard ratios (HRs) were determined by Cox proportional hazard regression and adjusted for the parameters that correlated with P2XR expression (pN stage, M status, UICC stage, L status). One-way ANOVA and the Holm–Sidak test were used to analyze results from cell culture experiments. Values of *p*   <  0.05 were considered statistically significant.

## 3. Results

### 3.1. Extracellular ATP Levels Are Increased in Bladder Cancer Cell Line Cultures of High Malignancy

While numerous studies in recent years have established pro-tumorigenic roles of extracellular ATP and P2X receptors in various cancers [8,17,20,21,22,23,24,25], comparable data on bladder cancers are scarce. Therefore, we first tested in a cell culture model with bladder cells of different grades of malignancy how ATP signaling affects bladder cancer cell proliferation. We used TRT-HU-1 cells, an hTERT-immortalized and non-transformed human urothelial cell line [33], RT4 cells, a low-grade, well-differentiated, non-invasive papillary transitional cell carcinoma cell line, and T24 cells, a transitional cell carcinoma cell line that was established from a bladder cancer patient with a high-grade, invasive urothelial carcinoma and that expresses various purinergic receptor subtypes [34,35]. ATP accumulation in the TME is driven by hypoxia, cell death, or ATP release from stimulated inflammatory cells [8]. In addition, tumor cells themselves may actively release ATP and maintain high ATP concentrations in their surroundings to fuel their own growth [19]. To test whether intrinsic surrounding ATP levels correlated with growth in bladder cancer cells, we compared ATP concentrations in the supernatants of T24 cell cultures with those of RT4 and TRT-HU-1 cells and determined the doubling times of each cell line. ATP concentrations were more than twice as high in supernatants of highly malignant T24 cells than in RT4 or TRT-HU-1 cell culture supernatants (Figure 1A). Consistent with a lower grade of malignancy, both of these cell lines grew significantly slower than T24 cells with doubling times of 21.37 ± 0.24 h (TRT-HU-1) and 23.25 ± 0.78 h (RT4), respectively, compared to 15.1 ± 0.02 h in T24 cells (*p* < 0.001; Figure 1B). These data suggest that ATP may indeed exert pro-tumorigenic effects on bladder cancer cells.

### 3.2. Extracellular ATP Promotes Proliferation of High-Grade T24 Cells through P2X Receptors

We and others have previously reported that P2X1R, P2X4R, and P2X7R have growth-promoting effects in different cancer cells in vitro and in vivo [19,36,37,38]. To test whether this also applies to bladder cancer cells, we treated T24 cells with specific receptor antagonists and assessed proliferation. Inhibition of P2X1R (with NF023), P2X4R (with 5-BDBD), or P2X7R (with A438079) significantly increased the doubling time from 15.31 ± 0.21 h in untreated cells to 17.04 ± 0.04, 17.14 ± 0.25, and 17.71 ± 0.09 h, respectively (*p* < 0.001; Figure 1C,D). Combined treatment with the P2X1R and P2X7R antagonists had a significantly stronger inhibitory effect than treatment with either antagonist alone (doubling time: 18.24 ± 0.25 h). In agreement with these findings, the general P2 receptor inhibitor suramin suppressed proliferation even further by more than 40% (doubling time: 21.70 ± 0.76 h). ATP (10 µM), the natural P2XR agonist, slightly increased proliferation (doubling time: 14.58 ± 0.53 h) while treatment with the ATP-hydrolyzing enzyme apyrase had the opposite effect. Similarly, CBX, a hemichannel blocker that inhibits cellular ATP release through pannexin-1 and thus interferes with autocrine purinergic feedback signaling [39], inhibited proliferation (Figure 1D). Taken together, these results support a growth-promoting role of autocrine P2XR signaling in T24 bladder cancer cells.

### 3.3. High P2X1 and High Combined P2X1/P2X7 Receptor Expression Scores Are Associated with Clinicopathological Indicators of Cancer Progression in MIBC Patients

The results of the in vitro experiments suggested that P2X1R, P2X4R, and P2X7R could be markers of bladder cancer growth and progression. To test this possibility, we next assessed the expression profiles of these receptors in MIBC and analyzed possible associations between expression intensity and clinicopathological characteristics. A total of 173 MIBC patients were included in the study. The mean age at the time of surgery was 67.1 (±9.0) years. Consistent with previous studies, the majority of patients were male (73.4%) [3]. During the course of the study, 58 (33.5%) patients received adjuvant chemotherapy and 43 (24.7%) patients were treated with adjuvant radiotherapy. Mean follow-up was 3.0 years, and 144 patients (83.2%) died during the follow-up period. P2X1R, P2X4R, and P2X7R expression was scored and classified as “high” or “low” as described above (Section 2.4). Membranous and cytoplasmic expression of all three receptor subtypes was detectable in tumor cells. In addition, tumor-infiltrating immune cells displayed a strong immunoreactivity for all receptors. Representative microphotographs are shown in Figure 2. We examined whether P2XR expression correlated with demographic or clinical (age, gender, adjuvant therapy) or histopathological (pT staging category, pN stage, UICC stage, M status, R status, L status, V status, and Pn status) characteristics. While there were no significant associations between the described variables and P2X4R or P2X7R expression, high P2X1R expression was significantly associated with lymphovascular invasion (L; *p* = 0.02), lymph node metastasis (pN; *p* = 0.03) and UICC stage (*p* = 0.01). In addition, high combined expression of P2X1R and P2X7R was significantly associated with lymphovascular invasion (L; *p* = 0.007), lymph node metastasis (pN; *p* = 0.007), distant metastasis (M; *p* = 0.046), and UICC stage (*p* = 0.001). Patient demographics, tumor characteristics, and their associations with P2XR expression are summarized in Table 1.

### 3.4. High P2X1 and High Combined P2X1/P2X7 Receptor Expression Scores Are Associated with Reduced Overall Survival

Median OS of all patients after initial surgery was 17.0 months. Median OS in patients with low P2X1R expression was 23.5 ± 45.6 months and significantly longer than in patients with high P2X1R expression (13.3 ± 44.1 months; Figure 3A). There was no significant difference in median OS between patients with low (16.6 ± 46.0 months) and high (17.8 ± 44.9 months) P2X4R expression (Figure 3B). Similarly, OS did not differ between patients with high or low P2X7R expression (17.6 ± 55.8 vs. 16.4 ± 37.5 months; Figure 3C). However, patients with a high combined P2X1R/P2X7R score showed a significant decrease in OS (12.7 ± 25.4 months) as compared to patients with a low P2X1R/P2X7R score (36.4 ± 56.0 months; Figure 3D).

### 3.5. High Expression of P2X1 and P2X7 Receptors Is Associated with Reduced Tumor-Specific Survival

Bladder-cancer-associated death occurred in 94 patients (54.34%,) and median TSS was 15.07 months. TSS in patients with high P2X1R expression (11.8 ± 10.0 months) was significantly lower than in patients with low P2X1R expression (16.4 ± 20.8 months; Figure 4A). P2X4R expression did not significantly affect TSS (Figure 4B). Patients with low P2X7R expression tended to have improved TSS compared to patients with high P2X7R expression, and this seemed to especially apply to patients who survived the first two years after diagnosis; however, the difference in median TSS between the two groups was not statistically significant (*p* = 0.051; Figure 4C). In contrast, a significant reduction in TSS was associated with tumors displaying a high P2X1R/P2X7R score when compared to tumors with a low P2X1R/P2X7R score (12.5 ± 9.8 months vs. 18.7 ± 22.5 months; Figure 4D).

### 3.6. High Combined Expression of P2X1 and P2X7 Receptors Is Associated with Reduced Disease-Free Survival

Median DFS of all patients was 11.4 months. It was not affected by the expression intensity of any single P2XR subtype (Figure 5A–C). However, there was a significant reduction in median DFS in patients with high combined P2X1R/P2X7R scores (8.7 ± 28.3 months) as compared to patients with low P2X1R/P2X7R scores (26.0 ± 57.8 months; Figure 5D).

### 3.7. Combined Expression of P2X1 and P2X7 Receptors Is an Independent Predictor of Overall and Tumor-Specific Survival

Multivariable survival analyses were calculated for OS, TSS, and DFS and adjusted for clinicopathological covariates that correlated with P2XR expression (pN stage, M status, L status, UICC stage; Table 1). A high combined P2X1R/P2X7R expression score independently predicted reduced OS (HR = 2.42; 95% confidence interval [CI]: 1.28–4.55; *p* = 0.006) and TSS (HR = 2.79; 95%-CI = 1.28–6.13; *p* = 0.01) in MIBC patients. For P2X1R, P2X4R, or P2X7R expression, no statistically independent influence on survival was found. The results of the multivariate analysis are summarized in Table 2.

## 4. Discussion

P2XRs are expressed by multiple malignant tumors and are increasingly recognized as prognostic indicators and potential therapeutic targets [8,12,40]. Here, we found that high expression of P2X1R alone or in combination with P2X7R predicts poor outcome in MIBC patients. P2X1R expression was associated with clinicopathological parameters of tumor progression and shorter median OS and TSS. In agreement, a recent study reported a link between high levels of P2X1R mRNA transcripts and shorter median OS in a cohort of bladder cancer patients from The Cancer Genome Atlas (TCGA) databank [34]. P2X1R expression was also previously implicated in the development of prostate cancer and acute pediatric leukemia [41,42].

We found that the combined P2X1R/P2X7R expression score increased the prognostic power as compared to P2X1R expression alone. High P2X1R/P2X7R scores were associated with reduced survival and an increased risk of distant metastasis. P2X7R is the most extensively studied P2XR subtype in the context of cancer (reviewed in [8,17,26,43,44,45]). Its role in cancer progression is complex. A distinctive feature of P2X7R is its ability to form a cell-permeabilizing macropore when stimulated by high concentrations of ATP (0.3–0.5 mM), which can induce apoptosis and cell death [46]. In agreement, decreased expression of P2X7R is associated with tumor progression in some cancers [45,47]. However, the fact that P2X7R is widely expressed in many tumors indicates that tumor cells have developed strategies, like the expression of P2X7R isoforms, to circumvent pore formation while maintaining the function of P2X7R as a Ca^2+^ membrane channel and stimulator of pro-proliferative metabolic pathways [8,36,37]. Accordingly, there are several studies demonstrating that P2X7R promotes tumor cell growth. For instance, P2X7R stimulation increases the proliferation of ovarian and pancreatic carcinoma, osteosarcoma, neuroblastoma, and leukemia cell lines [19,48,49,50,51,52,53]. Furthermore, P2X7R overexpression was linked to poor outcome in gastric, liver, lung, colorectal, and renal cell carcinoma [20,21,22,23,54]. Consequently, selective P2X7R antagonists have been shown to inhibit tumor growth and cancer cell migration and invasion in vitro and in vivo [37,38,48,55].

In our MIBC patient cohort, median OS and TSS were not significantly affected by P2X7R expression alone but long-term survival appeared to be negatively impacted by high expression levels of P2X7R. A pro-tumorigenic role of P2X7R in MIBC is further supported by our finding that high combined expression of P2X1R and P2X7R was significantly associated with lymph node and distant metastases and reduced OS, TSS, and DFS. Furthermore, as opposed to P2X1R expression alone, high P2X1R/P2X7R expression was an independent negative predictor of OS and TSS after adjusting for key clinicopathological parameters. In support of these results, we found that simultaneous inhibition of P2X1R and P2X7R impaired proliferation of the highly malignant bladder cancer cell line T24 more strongly than selective blockade of the individual receptors. Like most ionic receptors, P2XRs can form heteromers with other P2XR subunits, which typically differ in their pharmacological and functional properties from the respective homomers [56]. However, hetero-oligomerization of P2X7R has not been described so far [57]. It seems therefore likely that both P2X1R and P2X7R exert their tumor-growth-promoting effects in bladder cancer independently from each other in an additive way. We have previously found that in Jurkat cells, a lymphoblastic leukemia cell line, tonic ATP release, and autocrine stimulation of P2X1R and P2X7R increase Ca^2+^ influx and mitochondrial metabolism and promote proliferation [19]. These findings are consistent with substantial proof available for the trophic/growth-promoting effect of P2X7R that has been ascribed to its interaction with multiple pathways in the cellular energy metabolism [26,36,37,58]. The mechanisms behind the pro-tumorigenic effect of P2X1R stimulation are less well studied but might involve similar calcium-dependent pathways.

The results of our in vitro studies support the concept of a tonic growth-promoting stimulation of P2XRs in bladder cancer cells: Proliferation of T24 cells was not only promoted by stimulating cells with the natural P2XR agonist ATP. Proliferation was also impaired by inhibition of ATP release channels, degradation of cell-derived extracellular ATP by apyrase treatment, and blocking of purinergic receptors with antagonists, i.e., by interfering with autocrine purinergic signaling. Furthermore, we found that the highly malignant bladder cancer cell line T24 maintained higher surrounding ATP levels in the cell culture and grew significantly faster than bladder cancer cells of lower malignancy (RT24) or non-transformed immortalized TRT-HU-1 cells. This is consistent with reduced expression of ATP-converting ectonucleotidases and a reduction in the ATP-hydrolyzing capacity described in bladder cancer cells of high malignancy [59,60]. On the other hand, it was previously reported that high concentrations of ATP (1 mM) exert anti-proliferative effects on the highly malignant bladder cancer cell line HT-1376 [61]. The discrepancy in our findings can most likely be explained by cell-type specific differences and/or the use of higher ATP concentrations that are sufficient to induce macropore opening.

Only a few studies have investigated the role of P2X4R in cancer so far with sometimes contradictory results. Anti-proliferative effects of P2X4R were described in gastric and breast cancer cells in vitro [62,63]. On the other hand, P2X4R-mediated signaling processes enhanced tumor growth, invasion, and metastasis in breast cancer in vivo [64]. He et al. reported that P2X4R was the predominant P2 receptor in prostate carcinoma cells and demonstrated that inhibition of P2X4R impaired the growth and mobility of cancer cells [65]. In addition, a recently published study demonstrated that P2X4R-dependent signal transduction contributes to the survival of colon carcinoma cells under chemotherapy [66]. Even though we found that P2X4R blockade inhibited T24 cell proliferation in vitro, P2X4R expression was not significantly associated with survival in MIBC patients. One possible explanation is the influence of the tumor microenvironment and tumor-infiltrating lymphocytes on tumor progression in vivo. Purinergic signaling regulates immune cells in multiple ways [67]. The purinergic environment in the TME can promote both antitumor immunity and cancer immune evasion, depending on the expression profiles of purinergic receptors and other components of the purinergic signaling system, such as ectoenzymes, which determine ATP and adenosine concentrations in the TME. We focused in our study on P2X1R, P2X4R, and P2X7R, which are the predominant P2XRs in immune cells [27]. We found that these P2XRs are also expressed in tumor-infiltrating immune cells, with P2X4R in particular showing strong immunoreactivity. It is therefore likely that purinergic signaling affects not only cancer cell growth but also shapes the antitumor immune response against MIBC.

While we restricted our study to MIBC patients, who bear the poorest prognosis, most newly diagnosed bladder cancers are non-muscle-invasive bladder cancers (NMIBCs) [3,68]. NMIBC has a high recurrence rate, imposes a great psychological burden on patients, and is associated with high treatment costs [68]. An accurate risk classification is critical to avoid over- or undertreatment. Analogous to our findings in MIBC, P2X1R/P2X7R expression could be potentially useful in NMIBC as well to predict progression and guide individualized treatment decisions.

While we demonstrated that P2X1R/P2X7R expression is linked to lymph node and distant metastasis, the detailed underlying molecular mechanisms remain unclear. In addition, other purinergic receptor subtypes might affect malignancy factors such as tumor growth, tumor cell migration, invasion, and metastasis in different types of bladder cancer. Of note, high P2X6R expression was recently associated with prolonged survival in a mixed population of low- and high-grade bladder cancer patients [34]. Thus, further studies are needed to explore the potential prognostic and therapeutic value of the different P1 and P2 receptor subtypes in MIBC and NMIBC in more detail and define the role of purinergic signaling in bladder cancer development and progression as well as in the antitumor immune response.

## 5. Conclusions

The prognosis of advanced bladder cancer is poor and current therapies are often associated with severe side effects and significant costs. To avoid overtreatment, there is a high demand for new markers that can accurately assess the prognosis of patients with MIBC [67]. Our results suggest that P2X1R/P2X7R expression scores can be used as reliable and powerful prognostic markers. In addition, P2XR-mediated pathways may present potential targets for new innovative therapeutic strategies in MIBC. In particular, intravesical application of purinergic inhibitors could be explored as a new component of combinatorial therapies. However, further studies are first needed to characterize the role of purinergic signaling in bladder cancer growth and metastasis, as well as in the regulation of tumor-infiltrating immune cells, in more detail.

## Figures and Tables

**Figure 1 cancers-15-02321-f001:**
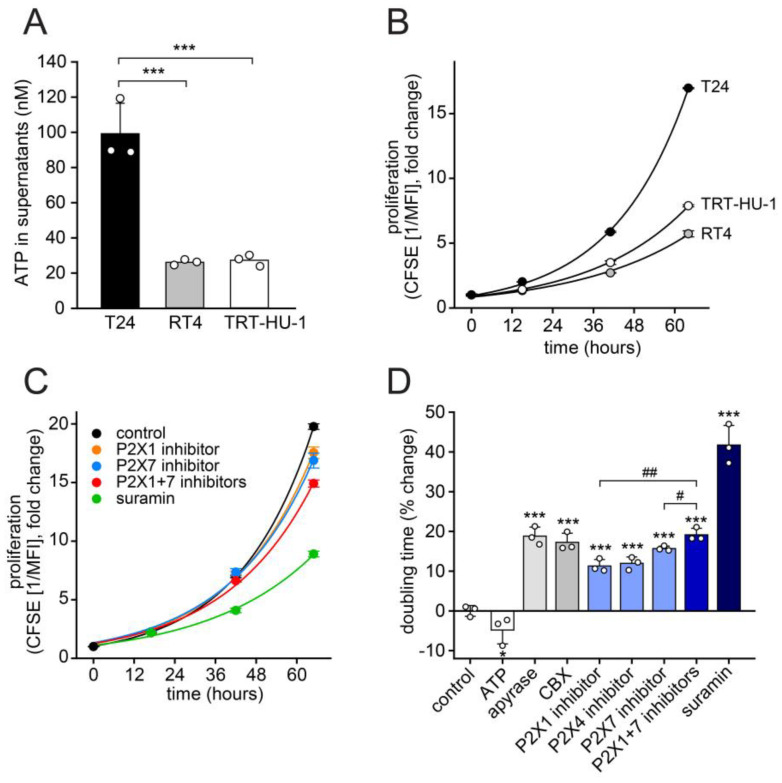
Extracellular ATP promotes proliferation of T24 cells through P2X receptors. (**A**) ATP concentrations in supernatants of T24, RT4, and TRT-HU-1 cell cultures were determined by HPLC (mean ± SD, *n* = 3 separate experiments; *** *p* < 0.001, one-way ANOVA). (**B**) T24, TRT-HU-1, and RT4 cells were labeled with CFSE and proliferation was determined at the indicated time points by flow cytometry (mean ± SD, *n* = 3 separate experiments). (**C**) T24 cells were treated with inhibitors of P2X1 (NF023; 10 µM) and/or P2X7 (A438079; 10 µM) receptors or with the general P2 receptor antagonist suramin (100 µM) and proliferation was assessed at the indicated time points (mean ± SD, *n* = 3 separate experiments). (**D**) Doubling times of T24 cells treated with P2 receptor inhibitors as in **C** or with ATP (10 µM), apyrase (1 U/ml), CBX (20 µM), or the P2X4R antagonist 5-BDBD (10 µM) were determined after 72 hours (mean ± SD, *n* = 3 separate experiments; * *p* < 0.05, *** *p* < 0.001 vs. control. # *p* < 0.05, ## *p* < 0.01, one-way ANOVA); MFI, mean fluorescence intensity.

**Figure 2 cancers-15-02321-f002:**
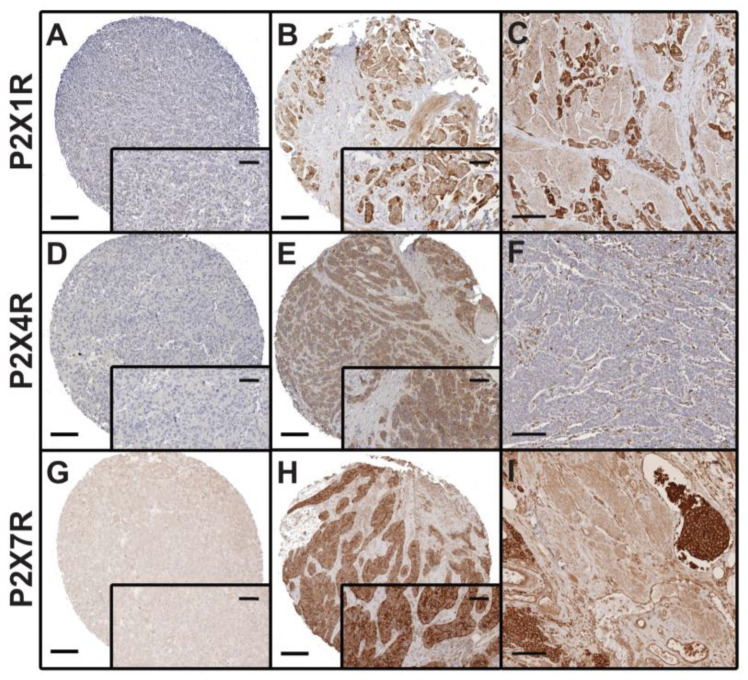
P2X1, P2X4, and P2X7 receptor expression in muscle-invasive bladder cancer (MIBC). (**A**–**C**) Representative photomicrographs show TMA samples with low (**A**) and high (**B**) P2X1R expression in cancer cells of MIBC. Compared to smooth muscle cells, muscle infiltrating cancer cells in (**C**) show strong distinct membranous staining. (**D**–**F**) TMA samples of MIBC with low (**D**) and high (**E**) P2X4R expression are shown. MIBC cells exhibit membranous and partially fine-granular cytoplasmic P2X4R staining, while tumor-infiltrating immune cells show strong immunoreactivity (**F**). (**G**–**I**) Low and high P2X7R expression patterns in MIBC are shown in (**G**) and (**H**), respectively. Cancer cells between muscle cells and in lymphatic vessels are highlighted by strong membranous P2X7R expression (**I**). Scale bars equal 100 μm (outer pictures and (**C**,**F**,**I**)) and 20 μm (inserts).

**Figure 3 cancers-15-02321-f003:**
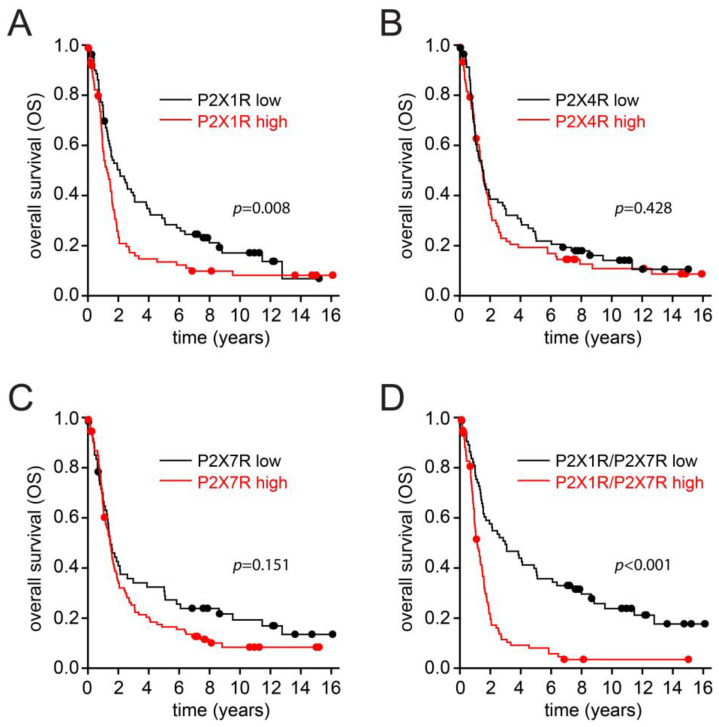
Effect of P2X receptor expression levels on overall survival in muscle-invasive bladder cancer (MIBC). Expression of P2X1R (**A**), P2X4R (**B**), P2X7R (**C**), and combined expression of P2X1R and P2X7R (**D**) was immunohistochemically analyzed in MIBC and scored as “low” (below median) or “high” (above median). Kaplan–Meier overall survival (OS) curves were plotted, and groups were compared by the log-rank test; (**A**) and (**B**): *n* = 171; (**C**): *n* = 172; (**D**): *n* = 170.

**Figure 4 cancers-15-02321-f004:**
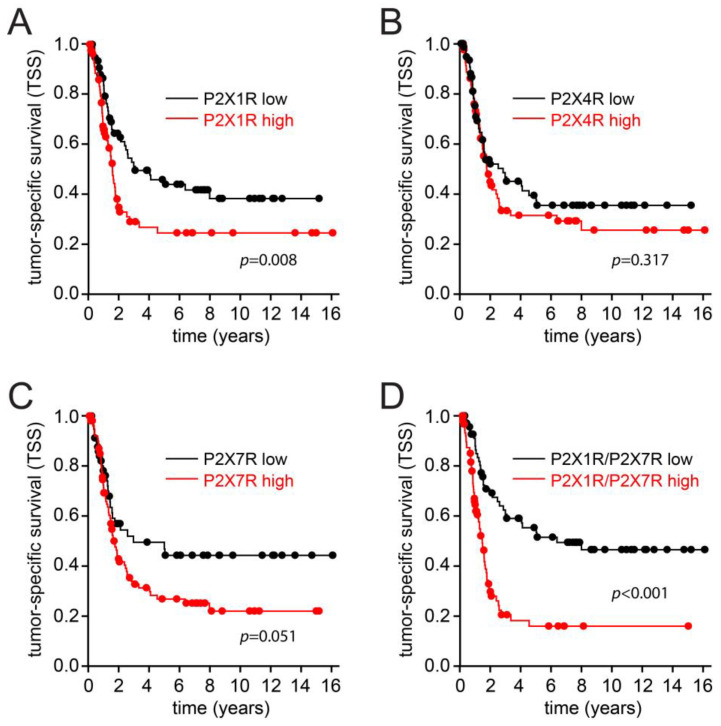
Effect of P2X receptor expression levels on tumor-specific survival (TSS) in muscle-invasive bladder cancer (MIBC). Expression of P2X1R (**A**), P2X4R (**B**), P2X7R (**C**), and combined expression of P2X1R and P2X7R (**D**) was immunohistochemically analyzed in MIBC and scored as “low” (below median) or “high” (above median). Kaplan–Meier tumor-specific survival (TSS) curves were plotted, and groups were compared by the log-rank test; (**A**,**B**): *n* = 171; (**C**): *n* = 172; (**D**): *n* = 170.

**Figure 5 cancers-15-02321-f005:**
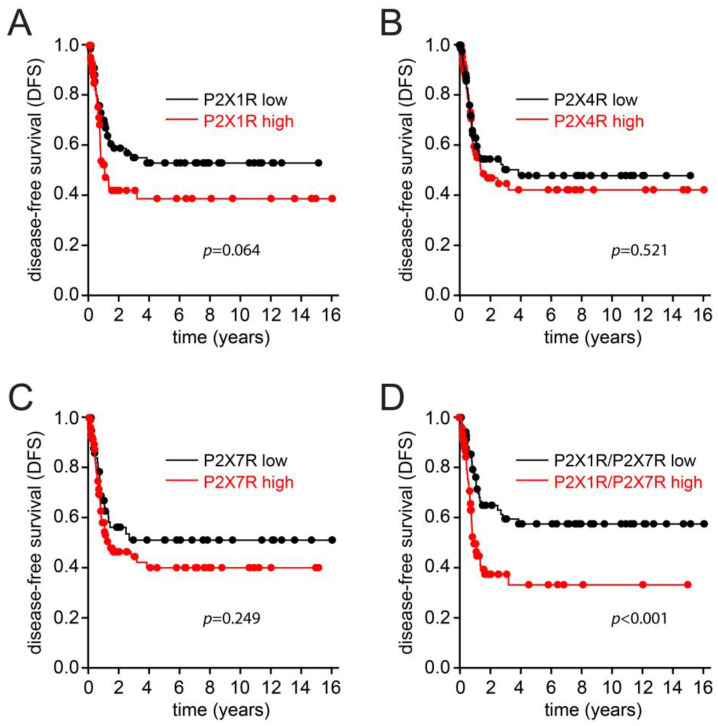
Effect of P2X receptor expression levels on disease-free survival (DFS) in muscle-invasive bladder cancer (MIBC). Expression of P2X1R (**A**), P2X4R (**B**), P2X7R (**C**), and combined expression of P2X1R and P2X7R (**D**) was immunohistochemically analyzed in MIBC and scored as “low” (below median) or “high” (above median). Kaplan–Meier disease-free survival (DFS) curves were plotted, and groups were compared by the log-rank test; (**A**,**B**): *n* = 171; (**C**): *n* = 172; (**D**): *n* = 170.

**Table 1 cancers-15-02321-t001:** Demographics, clinicopathological characteristics, and associations with P2X1, P2X4, and P2X7 receptor expression.

	Study	P2X1R (*n* = 171)			P2X4R (*n* = 171)			P2X7R (*n* = 172)			P2X1R/P2X7R, Combined (*n* = 170)		
	Population	Low	High	*p*	Low	High	*p*	Low	High	*p*	Low	High	*p*
	*n* = 173	*n* = 83	*n* = 88		*n* = 81	*n* = 90		*n* = 61	*n* = 111		*n* = 73	*n* = 97	
Age at diagnosis (years)													
Mean (SD)	67.1 (9.0)	66.8 (9.2)	67.5 (9.0)	0.67 *	67.3 (9.1)	67.0 (9.0)	0.83 *	67.0 (8.1)	67.1 (9.5)	0.95 *	67.0 (8.8)	67.3 (9.1)	0.83 *
Sex				0.98 ^#^			0.66 ^#^			0.87 ^#^			0.11 ^#^
Female (%)	46 (26.6)	22 (26.5)	22 (25.0)		20 (24.7)	24 (26.7)		16 (26.2)	29 (26.1)		14 (19.2)	30 (30.9)	
Male (%)	127 (73.4)	61 (73.5)	66 (75.0)		61 (75.3)	66 (73.3)		45 (73.8)	82 (73.9)		59 (80.8)	67 (69.1)	
T stage				0.55 ^#^			0.36 ^#^			0.21 ^#^			0.77 ^#^
T2 (%)	33 (19.1)	20 (24.1)	12 (13.6)		17 (21.0)	15 (16.7)		14 (23.0)	18 (16.2)		21 (28.8)	10 (10.3)	
T3 (%)	100 (57.8)	46 (55.4)	53 (60.2)		42 (51.9)	57 (63.3)		29 (47.5)	71 (64.0)		36 (49.3)	63 (65.0)	
T4 (%)	40 (23.1)	17 (20.5)	23 (26.1)		22 (27.2)	18 (20.0)		18 (29.5)	22 (19.8)		16 (21.9)	24 (24.7)	
Lymph node status				**0.03 ^#^**			0.99 ^#^			0.93 ^#^			**0.007 ^#^**
pN0 (%)	97 (56.1)	52 (62.7)	43 (48.9)		45 (55.6)	50 (55.6)		35 (57.4)	61 (55.0)		49 (67.1)	45 (46.4)	
pN+ (%)	61 (35.3)	22 (26.5)	39 (44.3)		28 (34.6)	33 (36.7)		21 (34.4)	40 (36.0)		18 (24.7)	43 (44.3)	
pNX (%)	15 (8.7)	9 (10.8)	6 (6.8)		8 (9.9)	7 (7.8)		5 (8.2)	10 (9.0)		6 (8.2)	9 (9.3)	
UICC stage				**0.01 ^#^**			0.35 ^#^			0.57 ^#^			**<0.001 ^#^**
2 (%)	27 (15.6)	19 (22.9)	7 (8.0)		15 (18.5)	11 (12.2)		11 (18.0)	15 (13.5)		20 (27.4)	5 (5.2)	
3 (%)	64 (37.0)	37 (44.6)	27 (30.7)		29 (35.8)	34 (37.8)		22 (36.1)	42 (37.8)		32 (43.8)	32 (33.0)	
4 (%)	82 (47.4)	27 (32.5)	54 (61.4)		37 (45.7)	45 (50)		28 (45.9)	54 (48.6)		21 (28.8)	60 (61.9)	
Vascular invasion				0.66 ^#^			0.92 ^#^			0.30 ^#^			0.87 ^#^
V0 (%)	145 (83.8)	68 (81.9)	75 (85.2)		67 (82.7)	76 (84.4)		54 (88.5)	90 (81.1)		60 (82.2)	82 (84.5)	
V1 (%)	28 (16.2)	15 (18.1)	13 (14.8)		14 (17.3)	14 (15.6)		7 (11.5)	21 (18.9)		13 (17.8)	15 (15.5)	
Lymphovascular invasion				**0.02 ^#^**			0.16 ^#^			1.00 ^#^			**0.007 ^#^**
L0 (%)	78 (45.1)	46 (55.4)	32 (36.4)		42 (51.9)	36 (40.0)		27 (44.3)	51 (45.9)		43 (58.9)	35 (36.1)	
L1 (%)	95 (54.9)	37 (44.6)	56 (63.6)		39 (48.1)	54 (60.0)		34 (55.7)	60 (54.1)		30 (41.1)	62 (63.9)	
Perineural invasion				0.68 ^#^			0.51 ^#^			1.00 ^#^			0.61 ^#^
Pn0 (%)	129 (74.6)	63 (75.9)	65 (73.9)		63 (77.8)	65 (72.2)		45 (73.8)	83 (74.8)		53 (72.6)	74 (76.3)	
Pn1 (%)	44 (25.4)	20 (24.1)	23 (26.1)		18 (22.2)	25 (27.8)		16 (26.2)	28 (25.2)		20 (27.4)	23 (23.7)	
Distant metastasis				0.14 ^#^			0.80 ^#^			0.34 ^#^			**0.046 ^#^**
M0 (%)	152 (87.9)	77 (92.8)	73 (83.0)		70 (86.4)	80 (88.9)		56 (91.8)	95 (85.6)		69 (94.5)	81 (83.5)	
M1 (%)	21 (12.1)	6 (7.2)	15 (17.0)		11 (13.6)	10 (11.1)		5 (8.2)	16 (14.4)		4 (5.5)	16 (16.5)	
Resection margin				0.17 ^#^			0.92 ^#^			0.35 ^#^			0.21 ^#^
R0 (%)	144 (83.2)	73 (88.0)	69 (78.4)		68 (84.0)	74 (82.2)		48 (78.7)	95 (85.6)		64 (87.7)	77 (79.4)	
R1 (%)	29 (16.8)	10 (12.0)	19 (21.6)		13 (16.0)	16 (17.8)		13 (21.3)	16 (14.4)		9 (12.3)	20 (20.6)	
Adjuvant Chemotherapy				0.96 ^#^			0.11 ^#^			0.98 ^#^			0.70 ^#^
Yes (%)	58 (33.5)	27 (32.5)	29 (33.0)		32 (39.5)	24 (26.7)		21 (34.4)	37 (33.3)		22 (30.1)	34 (35.1)	
No (%)	115 (66.5)	56 (67.5)	59 (67.0)		49 (60.5)	66 (73.3)		40 (65.6)	74 (66.7)		51 (69.9)	63 (64.9)	
Adjuvant Radiotherapy				0.82 ^#^			0.76 ^#^			0.78 ^#^			0.27 ^#^
Yes (%)	43 (24.7)	20 (24.1)	23 (26.1)		19 (23.5)	24 (26.7)		14 (23.0)	29 (26.1)		15 (20.5)	28 (28.9)	
No (%)	130 (75.3)	63 (75.9)	65 (73.9)		62 (76.5)	66 (73.3)		47 (77.0)	82 (73.9)		58 (79.5)	69 (71.1)	

* *t*-test; ^#^ Chi-square test; **bold font** indicates statistical significance; SD: standard deviation; P2XR: P2X receptor.

**Table 2 cancers-15-02321-t002:** Multivariate analysis of potential prognostic survival factors in MIBC patients.

			OS			TSS			DFS	
	*n* (%)	HR	95% CI	*p*	HR	95% CI	*p*	HR	95% CI	*p*
Lymph node status										
pN0	97 (61.4)	Reference								
pN+	61 (38.6)	0.98	0.59–1.65	0.95	0.88	0.47–1.66	0.69	3.41	1.83–6.36	**<0.001**
Distant Metastasis										
M0	152 (87.9)	Reference								
M1	21 (12.1)	1.73	1.02–2.91	**0.04**	2.03	1.11–3.72	**0.02**	2.19	1.18–4.64	**0.02**
UICC stage										
2	27 (15.6)	Reference								
3–4	146 (84.4)	6.53	3.02–14.14	**<0.001**	11.93	2.82–50.25	**<0.001**	5 × 10^8^	0–(+∞)	0.99
Lymphovascular invasion										
L0	78 (45.1)	Reference								
L1	95 (54.9)	1.56	0.93–2.62	0.09	1.65	0.86–3.13	0.13	8.62	4.04–18.36	**<0.001**
P2X1R (*n* = 171)										
Low	83 (48.5)	Reference								
High	88 (51.5)	0.75	0.43–1.33	0.33	0.71	0.35–1.42	0.33	0.80	0.36–1.88	0.60
P2X4R (*n* = 171)										
Low	81 (47.4)	Reference								
High	90 (52.6)	1.00	0.69–1.44	0.99	1.09	0.69–1.72	0.73	1.03	0.61–1.74	0.92
P2X7R (*n* = 172)										
Low	61 (35.5)	Reference								
High	111 (64.5)	0.84	0.56–1.28	0.42	0.89	0.52–1.51	0.66	0.72	0.41–1.27	0.25
P2X1R/P2X7R combined (*n* = 170)										
Low	73 (42.9)	Reference								
High	97 (57.1)	2.42	1.28–4.55	**0.006**	2.79	1.28–6.13	**0.01**	1.91	0.80–4.55	0.14

**Bold font** indicates statistical significance; OS: overall survival; TSS: tumor-specific survival; DFS: disease-free survival; HR: hazard ratio; CI: confidence interval; P2XR: P2X receptor.

## Data Availability

All data generated or analyzed during this study are included in the manuscript. Further inquiries should be directed to the corresponding author.

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
