# Peer review of "P2X1 and P2X7 Receptor Overexpression Is a Negative Predictor of Survival in Muscle-Invasive Bladder Cancer"

_cancers, 2023, doi:10.3390/cancers15082321_

Round 1

Reviewer 1 Report

The authors need to be congratulated for this interesting study. They present a well performed an well written study that covers an important topic in uro-oncology. 

The authors present a study on the role of ATP-receptor-expression (P2X1 , P2X7) as predictors for the oncologic outcome of patients with muscle invasive bladder cancers. Indeed, there is a need for reliable prognostic markers as the authors state. The receptor-over-expression is already used as a prognostic marker for other tumors but has not been tested for bladder cancer so far. This is the first study to do so. 

They have used tissue microarray, immunohistochemistry, semiqu. receptor expression analysis, cell cultures etc. Henve, they have looked deeply into the topic. 

They conclude that P2X1R/P2X7R expression scores can be used as reliable prognostic markers. This conculsion can be drawn from the presented results. 

Tables and Figures: good. 

This study can lead other researchers to further investigating this principle in urothelial cancer and therfore I expect it to be highly citable. 

I would suggest that the authors write a short note covering the potential benefit of P2X1R/P2X7R expression scores on non-msucle invasive bladder cancer in the discussion section of the manuscript. And I encourage the authors in a future study (not this study) to investigate the principle in non-muscel invasive bladder cancer, which potentially could lead patients in individualizing follow-up or treatment decisions. 

Author Response

Response:

We thank the reviewer very much for the positive evaluation of our study. It is an interesting and important question whether P2X1R/P2X7R expression scores can similarly predict outcome in non-muscle invasive bladder cancer and guide individualized treatment decisions in these patients. We thank the reviewer for this valuable comment. We briefly discuss this possibility in our revised manuscript.

Reviewer 2 Report

1. Please use the comma in the number in lines 47 and 48.

2. Does the text in line 50 mean metastasis?

3. The full name of CFSE should be indicated in the first appearance.

4. Do the authors use the P2X7R agonist in the studies? Is CBX the P2XR agonist?

5. Fig. 3.5 and 3.6 may be put in the same section.

6. The results in Table 1 show that overexpression of P2X2 and P2X7 are correlated with lymph node status, stage, lymphovascular invasion, and metastasis. Thus only these four factors may be included in multivariate analyses.

7. The expression of P2X2 and P2X7 is linked to metastasis. The amount of P2X2, P2X4, and P2X7 can be measured in T24, RT4, and TRT-HU-1 cells. The cell model of migration and invasion may be examined in this study to explore the molecular mechanism.

Author Response

Response:

We thank the reviewer for the careful evaluation of our manuscript and the useful suggestions that helped us improve our manuscript.

  1. Please use the comma in the number in lines 47 and 48.

Thank you for pointing out this mistake, we corrected it.

  1. Does the text in line 50 mean metastasis?

Thank you for this question. The expression “beyond the bladder” refers to stage III or IV disease, i.e., cancer growth into surrounding tissues and/or metastasis. We revised the text to clarify this point.

  1. The full name of CFSE should be indicated in the first appearance.

Thank you for this comment, we added the full name.

  1. Do the authors use the P2X7R agonist in the studies? Is CBX the P2XR agonist?

We used ATP, which is the natural agonist of P2X7R (and all other P2X receptors) as there is no receptor-specific agonist available. CBX (carbenoxolone) is a blocker of hemichannels like, e.g., pannexin-1, which is involved in active cellular ATP release in many cell types, including cancer cells. We modified respective passages in the Methods and Results sections to better clarify the roles of the different agents we used.

  1. 3.5 and 3.6 may be put in the same section.

We agree that these data are related to each other and could as well be presented together in one section. For reasons of consistency, we decided to present each of the three survival figures (Fig. 3, 4, 5) in separate sections.

The results in Table 1 show that overexpression of P2X2 and P2X7 are correlated with lymph node status, stage, lymphovascular invasion, and metastasis. Thus only these four factors may be included in multivariate analyses.

Thank you for pointing this out. We adjusted the factors included in the multivariate analysis as recommended and revised Table 2 accordingly.

  1. The expression of P2X2 and P2X7 is linked to metastasis. The amount of P2X2, P2X4, and P2X7 can be measured in T24, RT4, and TRT-HU-1 cells. The cell model of migration and invasion may be examined in this study to explore the molecular mechanism.

We thank the reviewer for these valuable suggestions. Besides growth behavior, invasion, migration and metastasis are certainly crucial factors that determine the malignancy of a cancer. We fully agree that it would be desirable and promising to evaluate these functions and the role of purinergic receptors in more detail. We believe that it is beyond the scope of the present study but emphasize in the revised Discussion the need to further investigate this important aspect.     

Reviewer 3 Report

Although the authors have done good job by studying the role of P2XRs in bladder cancer cell and the prognostic value of P2XR expression in MIBC patients.

However the manuscript would be better and comprehensive by addressing the following comments.

1. Human bladder(cancer) expresses seven P2XRs(P2X1—P2X7). What is the rationale of selecting P2X1R, P2X4R and P2X7R only. Have authors tried to check expression of other P2XRs in this study.

2. Validation method for the expression would make the study better.

3. Authors have not mentioned the controls used in the expression study of P2XR2.

4. Final English and Scientific language needs to be checked.

Author Response

Response:

We thank the reviewer for the overall positive evaluation of our manuscript and appreciate the meaningful suggestions.

  1. Human bladder(cancer) expresses seven P2XRs(P2X1—P2X7). What is the rationale of selecting P2X1R, P2X4R and P2X7R only. Have authors tried to check expression of other P2XRs in this study.

Thank you for this important question. P2X1R and P2X7R belong to the best studied P2XRs in cancer, and P2X1R, P2X4R and P2X7R have been linked to cancer progression in various tumors. Furthermore, P2X1R, P2X4R and P2X7R are the predominant P2XRs in immune cells. While we did not focus on anti-tumor immunity in our study, this is an important aspect that needs to be taken into account, especially when considering therapeutic applications. In addition, there are specific receptor antagonists available for these three but not the other P2XR subtypes, which facilitates their study in vitro and may also have implications for potential future therapeutic applications. We explain the rationale for our selection at the end of the Introduction section and in the Discussion. We agree, however, that it would be desirable to study the expression of all seven P2X receptor subtypes. We revised the Discussion to emphasize this point and encourage future studies to elucidate the roles of other purinergic receptors in bladder cancer.

  1. Validation method for the expression would make the study better.

Thank you for this comment. The manufacturers validated the staining against Western blot and RNA-seq data on a wide range of normal or cancer tissue. In addition, all antibodies have been successfully used in different studies before [1-3]. We included this information in the Methods section.

  1. George J, Cunha RA, Mulle C, Amedee T: Microglia-derived purines modulate mossy fibre synaptic transmission and plasticity through P2X4 and A1 receptors. The European journal of neuroscience 2016, 43(10):1366-1378.
  2. Johansson KE, Stahl AL, Arvidsson I, Loos S, Tontanahal A, Rebetz J, Chromek M, Kristoffersson AC, Johannes L, Karpman D: Shiga toxin signals via ATP and its effect is blocked by purinergic receptor antagonism. Scientific reports 2019, 9(1):14362.
  3. Walenta L, Fleck D, Frohlich T, von Eysmondt H, Arnold GJ, Spehr J, Schwarzer JU, Kohn FM, Spehr M, Mayerhofer A: ATP-mediated Events in Peritubular Cells Contribute to Sterile Testicular Inflammation. Scientific reports 2018, 8(1):1431.
  4. Authors have not mentioned the controls used in the expression study of P2XR2.

Thank you for pointing out this shortcoming. Antibody staining was established by default with appropriate isotype and system controls. Tonsil tissue was used as a positive control and was included in each staining run. We added this information to the Methods section.

  1. Final English and Scientific language needs to be checked.

Thank you very much for this suggestion. We carefully reviewed the text and made several edits to improve language quality and readability.

Round 2

Reviewer 2 Report

No comments